



**POTENTIAL IMPACTS OF CABLE BACTERIA ACTIVITY ON HARD-**
**SHELLED BENTHIC FORAMINIFERA: A PRELUDE TO IMPLICATIONS**
**FOR THEIR INTERPRETATION AS BIOINDICATORS OR PALEOPROXIES**
*Maxime Daviray[1]\*, Emmanuelle Geslin[1], Nils Risgaard-Petersen[2], Vincent V. Scholz[3], Marie Fouet[1,4],*
*Edouard Metzger[1]*
\*Correspondence: Maxime DAVIRAY (maxime.daviray@univ-angers.fr)
[1.] UMR CNRS 6112 - LPG, **Université d'Angers**, Nantes Université, Le Mans Université, CNRS, 49000 Angers, France
[2.] Aquatic Biology, Department of Biology, Aarhus University, 8000 Aarhus C, Denmark
[3.] Center for Electromicrobiology, Department of Biology, Aarhus University, 8000 Aarhus C, Denmark
[4.] at present: UMR CNRS 5805 EPOC - OASU, Station Marine d'Arcachon, Université de Bordeaux, CNRS, 33120
Arcachon, France

## 12   ABSTRACT

Cable bacteria (CB) are filamentous bacteria coupling sulphide oxidation to oxygen
reduction over centimetre distances. This bacterial activity generates a strong pH gradient
within the first few centimetres of the sediment that affects the microhabitats occupied by
benthic foraminifera. Hard-shelled foraminifera are protists able to build a calcareous or
agglutinated shell (called "test"). Here we study the impact of sediment acidification induced
by CB activity (CBA) on calcareous test preservation. For this study, sediment cores were
sampled in the macrotidal Auray estuary located on the French Atlantic coast. Living and dead
foraminifera were quantified (until 5-cm depth) and discriminated using the Cell-Tracker[TM]
Green vital marker. CBA was assessed with pH and oxygen profiles combined with quantitative
Polymerase Chain Reaction (q-PCR). On two different intertidal mudflats, volumetric filament
densities have been measured. They were comparable to those observed in the literature for
coastal environments, with $7.4 \pm 0.4$ and $74.4 \pm 5.0$ m.cm$^{-3}$ per bulk sediment respectively.
Highly contrasting CBA (from low to very intense) were described with lowest pH at 5.8. This
seems to lead to various dissolution stages of the foraminiferal calcareous test from intact to
fully dissolved tests revealing the organic lining. The dissolution scale is based on observations
of living *Ammonia* spp. and *Haynesina germanica* specimens under a Scanning Electronic
Microscope. Furthermore, dead foraminiferal assemblages showed a strong calcareous test
loss and an organic lining accumulation throughout depth under low pH, hampering the test
preservation in deep sediment. These changes in both living and dead foraminiferal
assemblages imply that CB must be strongly considered in ecological monitoring and historical
studies using foraminifera as bioindicator and paleoenvironmental proxy.



## 1 INTRODUCTION


Cable bacteria (CB) were discovered by Pfeffer and co-workers in 2012. They are sulphide-
oxidizing filamentous multicellular procaryotes from the Desulfobulbaceae family. They live in
marine and freshwater sediments all around the world (Risgaard-Petersen et al., 2015; Burdorf
et al., 2017). They inhabit a several centimetres thick zone from the oxic surface to the deep
sulphidic sediment. CB generate a vertical bioelectrical current by coupling the cathodic
oxygen or nitrate reduction at the sediment surface to the anodic sulphide oxidation at depth
(Nielsen et al., 2010; Pfeffer et al., 2012; Risgaard-Petersen et al., 2012; Marzocchi et al.,
2014). CB activity (CBA) strongly affects sediment geochemistry and results in a clear
geochemical fingerprint: an oxygen decrease in the surface sediments combined with a pH
maximum in this oxic zone, followed by a strong acidification of the pore water in the suboxic
zone (Nielsen et al., 2010; Risgaard-Petersen et al., 2012, 2014; Meysman et al., 2015). It
leads to iron sulphide and carbonate dissolution from the suboxic zone (Risgaard-Petersen et
al., 2012; Rao et al., 2016; van de Velde et al., 2016) and possibly the calcareous shell of
benthic organisms.
Benthic foraminifera are unicellular meiofaunal organisms. Most species can build a
hard-shell (called a test) that can be agglutinated (cemented grains), hyaline calcareous
(calcium carbonate) and porcelaneous calcareous (calcium carbonate enriched in
magnesium). Benthic foraminifera are very abundant in marine areas (Martin, 2000) including
transitional environments (Alve & Murray, 1999; Debenay et al., 2006). These systems located
between marine and continental areas (i.e. littoral and estuarine zones), are subjected to a
high variability of environmental factors (e. g. tide, freshwater flows, evaporation, development
of seagrass meadows over seasonal cycles...). Then, benthic foraminifera are submitted to
strong variability of physical and geochemical parameters such as temperature, salinity or pH
that they must tolerate. Despite such variability, benthic foraminifera assemblages have been
used in transitional environments as bioindicators for biomonitoring ecological state (Mojtahid
et al., 2006; Balsamo et al., 2012; O'Brien et al., 2021; Fouet et al., 2022) and as
paleoenvironmental proxies to understand past ecosystems functioning (Martin, 2000; Murray,
2006; Katz et al., 2010; Keul et al., 2017; Durand et al., 2018). However, species with a
calcareous test can be affected by low pH and carbonate undersaturation leading to test
dissolution (Le Cadre et al., 2003; Bentov et al., 2009; de Nooijer et al., 2009; Haynert et al.,
2011, 2014; Kurtarkar et al., 2011; Charrieau et al., 2018b). Even if they are rarely observed
*in situ*, few studies have reported signs of severe test dissolution in living assemblages (e.g.,
Alve and Nagy, 1986; Buzas-Stephens, 2005; Polovodova and Schonfeld, 2008; Haynert et
al., 2012; Cesbron et al., 2016; Charrieau et al., 2018a; Schönfeld and Mendes, 2022). These
authors attribute these dissolution observations to low pH and undersaturation of the carbonate



system, which would be due to abiotic conditions (anthropogenic pollution, freshwater
intrusions) or more rarely to biotic ones (degradation of plants). Under laboratory conditions,
Le Cadre et al (2003) have shown that test dissolution of living *Ammonia becarri* starts at pH

7.5.

Benthic foraminifera live mainly in the topmost sediment. CB develop also on the few

topmost centimetres of the sediment, which can therefore lead to an environmental overlap of
the bacterial and foraminiferal communities. Richirt et al 2022 hypothesised that CBA induces
the dissolution of calcareous tests within the sediment of the Lake Grevelingen (Netherlands).
In the present study, we assess the impact of cable bacteria activity on the foraminiferal test
preservation in sediment, testing the hypothesis that CBA is responsible for depleting the
preservation of calcareous foraminifera in benthic assemblages. To achieve this, CBA was
characterized by oxygen and pH microprofiling and CB density quantified by qPCR on intertidal
mudflats of the Auray estuary (French Atlantic coast). Calcareous test dissolution stages were
defined and quantified thanks to the analyse of SEM images. Then, we described living and
dead foraminiferal assemblages to assess the calcareous test loss.



## 2 MATERIALS AND METHODS

### 2.1 Studied Area

The Gulf of Morbihan (Atlantic coast, France) is an enclosed marine bay where the Auray river flows. The Auray estuary is a macrotidal estuary with a tide range about 4 m (**Figure 1**).

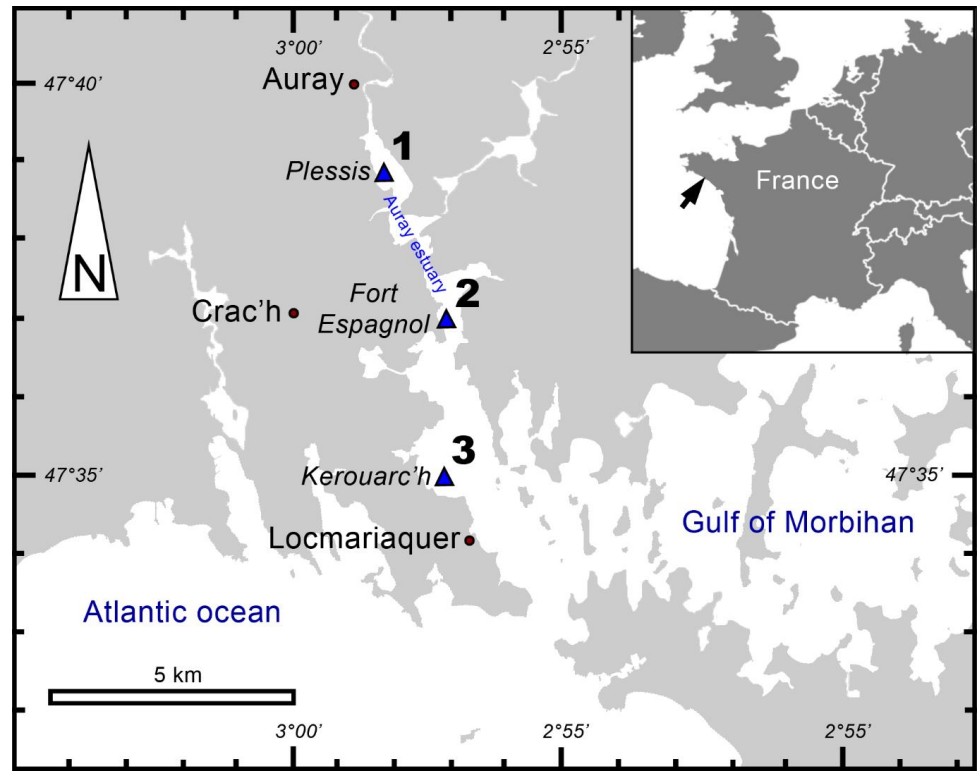

*Figure 1. Locations of sampling stations in the intertidal mudflats of the Auray estuary (France).*

Saltwater flows upstream over 20 km from the mouth of the estuary (357 m wide) which is tide-dominated (online data from OFB and IFREMER, accessed on May 05th 2022). The extensive description of this area was made by Fouet et al. (2022).

In September 2020, three stations along the Auray estuary were sampled on intertidal mudflats at low tide (**Figure 1** and **Table 1**): station 1 (Plessis), station 2 (Fort Espagnol) and stations 3 (Kerouarc'h). Characteristics of the sampled stations are presented in **Table 1**.

***Table 1. Characteristics of the stations sampled in September 2020.** Temperature and salinity values correspond to these measured on the sampling day; weighted average and SD of sediment density data from a previous campaign in 2019; (\*) name of the station after Fouet et al., (2022).*

| STATION | COORDINATES | DISTANCE FROM SEA | T (°C) | SALINITY | SEDIMENT DENSITY (g.cm⁻³) | VEGETATION COVER |
|---|---|---|---|---|---|---|
| 1 | 47.646° N, | 12 km | 24.4 | 29.6 | 1.71 ± 0.12 | *Ulvea* mat |



| | | | | | |
|---|---|---|---|---|---|
| (*6B) | -2.972° W | | | | |
| 2 (*4B) | 47.616° N, -2.953° W | 8 km | 21.2 | 38.5 | 1.67 ± 0.33 | *Ulvea* mat |
| 3 (*2C) | 47.583° N, -2.955° W | 4.3 km | 21.5 | 34.3 | 1.51 ± 0.23 | thick *Ulvea* mat few *Zostera* |

## 2.2 Sediment Sampling and Processing

One core was sampled from each station by hand with a Plexiglas® tube (82 mm inner
diameter, 50 mm depth) and was transported within an hour in a cool box to the field laboratory.
Then, the cores were submerged in ambient seawater for at least two hours to retrieve *in situ*
conditions before microprofiling.
After microprofiling, each core was sliced using a core pusher and two trowels. Slice
thickness was 2 mm for the first 20 mm depth, and 10 mm up to 50 mm depth. Each sediment
slice was treated with Cell-Tracker™ Green (CTG 5 CMFDA: 5-chloromethylfluorescein
diacetate; Molecular Probes, Invitrogen Detection Technologies) to mark living benthic
foraminifera by fluorescence (Bernhard and Bowser, 1996; Bernhard et al., 2006). One mg of
CTG was dissolved in 1 mL of dimethylsulfoxide (DMSO). This solution was then pipetted into
the flask containing the sediment slice and its volume of ambient water to get a final solution
of CTG about 1 µM (Bernhard et al., 2006; Pucci et al., 2009; Langlet et al., 2013, 2014;
Cesbron et al., 2016). Each sample was then incubated in dark at room temperature overnight
and then fixed with ethanol 99% (Choquel et al., 2021). Eventually, the samples were sieved
with tap water over 315-, 150-, 125- and 63-µm mesh screens. Samples were conserved in
99% ethanol.
DNA was extracted from sub-samples of sediment slices at stations 1 and 2. 1-2 g
every second slice down to 18-mm depth were sampled with a heat-sterilized spatula and
transferred to 2 ml Eppendorf tubes, then frozen at -20°C degrees. Samples were sent in dry
ice ($CO_{2(s)}$ at -50°C) to the Microbiology Institute of Biology in Aarhus University (Denmark) for
qPCR analysis to quantify cable bacteria biomass.

## 2.3 Microsensor Profiling

Two Unisense© profiling systems were used simultaneously. One consisted of two oxygen
Clark-type microsensors with a 50 µm tip (Revsbech and Jørgensen, 1986; Revsbech, 1989),
and the other of a pH sensor with a 500 µm tip diameter (PH500, Unisense). They were both
mounted on a motorized micromanipulator linked to a computer, and connected to a MultiMeter
S/N. The increment was 50 µm until 3 mm for oxygen. It was 100 µm around the seawater-
sediment interface (SWI) for pH, and it was adapted in real time according to the evolution of
the observed profile until 50 mm depth. For each core, eight descents were managed for $O_2$,





for a total of 16 profiles, while only one profiling was done for pH. To calibrate the $O_2$
microsensor, two points were measured, with the 100% of oxygen saturation in the bubbling
seawater column, and the 0% into the anoxic part of sediment. To calibrate the pH
microsensor, 3 NBS buffers were used (values 4.0, 7.0, 9.2).

## 132    2.4    Living Foraminiferal Analyses

Counts of hard-shell benthic foraminifera were performed in wet conditions (water) on the >125
µm fractions using an epifluorescence stereomicroscope (Olympus SZX12 with a light source
CoolLED $p$E-100, emission wavelength λ = 470 nm). All specimens showing clear green
fluorescence were picked and identified. Remaining specimens were considered as dead. In
doubtful cases, particularly with agglutinated species, specimens were crushed to inspect
whether fluorescence was due to the presence of protoplasm, to the autofluorescence of
sediment grains composing the test, or the presence of bacteria or nematodes living inside
(Langlet et al., 2013; Cesbron et al., 2016). Total foraminiferal densities were expressed per
50 $cm^2$ of sediment and foraminiferal densities for sediment layers per 10 $cm^3$ volume.
For the taxonomy of hard-shell foraminifera species, reference publications on
estuarine foraminifera (Feyling-Hanssen et al, 1972; Hansen et al, 1976; Murray et al, 1979;
Scott et al, 1980; Hayward et al, 2004; Schweizer et al, 2011; Camacho et al, 2015; Richirt et
al, 2019; Fouet et al., 2022; Jorissen et al., 2023), and the World Register of Marine Species
were used. The distinction between the *Ammonia* phylotypes (Richirt et al, 2019) being difficult,
on particular on the dissolved tests, the results will be discussed at the genus level.

## 148    2.5    SEM Imaging

Living foraminifera from three layers (0-2 / 6-8 / 40-50 mm depth), according to main pH
features, were all observed under a Scanning Electronic Microscope (SEM). Two different
high-resolution SEM were used: a DEBEN Hitachi TM4000 at the LPG (samples not
metallised, 15kV, wd = 6,5 mm, partial vacuum (60 Pa)) and a Zeiss EVO LS10 at the Service
Commun d'Imageries et d'Analyses Microscopiques of Angers University (SCIAM; samples
not metallised, 20 kV, wd = 6,5 mm, partial vacuum (60 Pa), amperage 200 to 250 pA). Few
scales of calcareous test dissolution of living foraminifera have been proposed in the literature
(Corliss and Honjo, 1981; Le Cadre, 2003b; Haynert et al., 2011; Gonzales et al., 2017;
Charrieau et al., 2018c; Schönfeld and Mendes, 2022). These authors proposed scales varying
from 4 to 5 different stages based on SEM images or stereomicroscope observations. They
used a wide variety of morphological criteria to describe each dissolution stage (i.e. the number
of calcite layers altered and chambers damaged, the presence of cracks or holes, whether the
inner organic lining was visible, etc.). In the present study, we propose a scale of six dissolution



stages based on SEM pictures of the two most abundant calcareous species in our living
assemblages (*Ammonia* spp. and *Haynesina germanica*).
***Table 2. Description of the six dissolution stages of the calcareous tests of Ammonia spp. and Haynesina***
***germanica.***

| DISSOLUTION STAGE | NAME | SEM OBSERVATIONS AND STAGE DESCRIPTIONS | FIGURES |
|---|---|---|---|
| DS-0 | Intact test | intact, glassy test with a smooth surface and cylindrical pores, no sign of dissolution. | **Fig. 3-1** **Fig. 4-1** |
| DS-1 | Slight surface dissolved test | transparent test with cylindrical pores, alteration of the last calcite layer only, appearance of the interpore sutures in *H. germanica* (scarce in *Ammonia* spp., alteration more visible on the inter-chamber walls). | **Fig. 3-2** **Fig. 4-2** |
| DS-2 | Peeled test | dull, whitish test with some fusion of adjacent widen pores, calcite layers cracking and crumbling, last chamber often lost, thinner and blunt tubercular ornamentation of *H. germanica*. | **Fig. 3-3** **Fig. 4-3** |
| DS-3 | Cracked test | opaque and cracked test with a strong alteration of all calcite layers, brittle test with holes, fusion of widen pores, the organic lining can be visible, loos of last chamber, broken ornamentation of *H. germanica*. | **Fig. 3-4** **Fig. 4-4** |
| DS-4 | "Star-shape" test | nearly completely dissolved test, only the inter-chamber walls remaining, the last chambers often absent, dissolved peripheral chambers with the inner organic lining visible. | **Fig. 3-5** **Fig. 4-5** |
| DS-5 | Fully dissolved test | totally dissolved test revealing the inner organic lining, may keep the foraminifera shape allowing the identification of the genus *Ammonia* (not observed for *H. germanica*). | **Fig. 3-6** |



**Figure 2. Dissolution scale of Ammonia spp. based on high-resolution SEM images (spiral view).** *The specimens are classified into six stages of test dissolution from intact (stage 0) to fully dissolved (stage 5). For stages 0 to 2, a zoom on the last formed chamber was done (**1-b, 2-b, 3-b**), and on the n-1 chamber for stage 3 (**4-b**). White arrows point the organic lining.*






*Figure 3. Dissolution scale of Haynesina germanica based on high-resolution SEM images.* *The specimens are classified into five stages of test dissolution from intact (stage 0) to the « star shape » (stage 4). No organic lining (stage 5) has been identified as belonging to the taxa Haynesina. For stages 0 and 1, a zoom on the last formed chamber was done (**1-b, 2-b**), and on the n-1 chamber for stages 2 and 3 (**3-b, 4-b**). White arrow points the organic lining.*




## 2.6 Dead Foraminiferal Analyses


Non fluorescent tests of foraminifera were counted as dead specimens and picked in wet
conditions (water) to preserve the organic linings from fully dissolved tests. We proceeded
under a stereomicroscope (ZEISS Stemi sv11) in three sediment layers: the surface layer (0-
2 mm), the subsurface layer (6-8 mm) and the deep layer (40-50 mm). After quick observations,
when high densities were estimated (above 500 individuals; Patterson and Fishbein, (1989))
fractions were splitted into 8 sub-samples using a wet splitter (Charrieau et al., 2018a).

## 2.7 Ratios in Foraminiferal Assemblages


In order to characterize the loss of calcareous in the assemblages, we defined a ratio as
follows:
$$C/T = calcareous\ foraminifera/total\ foraminifera$$
Calcareous foraminifera are counted regardless their dissolution stage and total
foraminifera include agglutinated individuals. To estimate the intensity of dissolution in the
assemblage, we calculated the following ratio:
$$DS\text{-}5/C = calcareous\ test\ at\ dissolution\ stage\ 5/total\ calcareous\ foraminifera$$
These ratios were calculated on both living and dead assemblages for layers 0-2 / 6-8
and 40-50 mm.

## 2.8 Statistical Procedure


The putative relationship between CBA and the advanced dissolution stages of the living
calcareous test foraminifera was assessed by applying the parametric Fisher's test followed
by the pair-wise Fisher's test for *post-hoc* comparisons were used. To minimize the risk type 1
error *p*-values were FDR-adjusted. The significance level was set to 5 %. As the last layer of
calcite produced during the growth of the foraminifera covers the entire test and is thinner than
the others (Haynes, 1981; Hansen, 1999; Debenay et al., 2000; Boudagher-Fadel, 2018), DS-
1 and 2 are more commonly observed resulting from a process of gradual dissolution or
precipitation of calcite. Discrimination of the effect of the dissolution process is therefore made
on the alteration of several calcite layers as for DS-3 and above. For this purpose, the
dissolution stages were combined into two groups: no to slight dissolution (DS-0, 1 and 2) and
moderate to severe dissolution (DS-3, 4 and 5). These two groups were then compared
between each of the three stations, and between the different depth levels (0-2 / 6-8 / 40-50
mm depth) for each station. Statistics were carried out using the *R* software using the "stats"
and "rstatix" packages.





## 2.9  Sediment Treatment for DNA Extraction and Quantification

DNA was extracted from weighed amounts of sediment (0.22 - 0.25 g wet weight). DNA extraction was carried out using DNeasy PowerLyzer PowerSoil Kit (Qiagen) and the DNA was collected in 60 µl elution buffer. The analysis followed the procedures outlined in Geelhoed et al. (2020). The primer combination of ELF645wF and CB836wR was used to target the 16S rRNA gene of the marine cable bacteria of the genus *Candidatus* Electrothrix Trojan, 2016. The calibration curves were obtained using a synthetic standard (sequence accession KR912339.1, position 611-912, synthesized by Eurofins Genomics, Denmark) diluted in a 10-fold dilution series. The standards and samples were run in triplicates. Each reaction contained the master mix (RealQ Plus 2x Master Mix Green, Low ROXTM, Ampliqon, Denmark), forward and reverse primers (0.2 µM), BSA (1 µM). The qPCR was performed with a real time PCR analyser (AriaMX, Agilent). The thermal cycles were as follows: 15 min at 95 °C for initial denaturation followed by 40 cycles of 15 s at 95 °C (denaturation), 30 s at 60 °C (annealing), and 20 s at 72 °C (amplification). Afterwards, the melting curve was obtained by 30 s at 95°C, 30 s at 60 °C, and 30 s at 95 °C. Finally, the temperature was held for 5 min at 40 °C to terminate the analysis. The results are reported as the unit gene copies.(g wet sediment)$^{-1}$. CB filament density were calculated as in Geelhoed et al. (2020), using data of wet sediment density from a previous campaign in 2019 (**Table 1**), and expressed in m.cm$^{-3}$.

## 3  RESULTS

### 3.1  Microsensor Profiles and Cable Bacteria Abundance

Oxygen penetration depth in the sediment at stations 1, 2 and 3 was 1.4 ± 0.2, 0.9 ± 0.3 and 0.9 ± 0.2 mm, respectively. At station 1, pH rapidly decreased from 7.7 at the Sea Water Interface (SWI) to a minimum of 6.8 at 15 mm depth. Below this minimum, pH stabilised to 7.2 around 40 mm depth. In contrast, at stations 2 and 3, pH increased immediately below the SWI from 7.8 to 8.1 at 0.8 mm depth and to 7.95 at 0.6 mm, respectively (**Figure 4**). Below these maxima, at both stations, pH reached a minimum of 5.8 at 7 mm depth at station 2 and of 6.3 between 7-19 mm depth at station 3. Below these minima, pH stabilised at 6.8 after 25 mm depth at station 2, and at 6.9 after 34 mm depth at station 3. Those profiles with an oxygen decrease in the surface sediments combined with a pH maximum in this oxic zone, followed by a strong acidification of the pore water in the suboxic zone, are typical CBA fingerprints.

At station 1, the number of 16S CB copies of *Candidatus* Electrothrix ranged from 0.23 ± 0.01 $_x$10$^7$ to 0.48 ± 0.01 $_x$10$^7$ 16S copies.(g wet sediment)$^{-1}$, and remained constant through depth (**Figure 4**). At station 2, it amounted to 2.8 ± 0.12 $_x$10$^7$ 16S copies.(g wet sediment)$^{-1}$ in the upper 5 mm of sediment and progressively decreased to about 0.3 ± 0.01 $_x$10$^7$ 16S





copies.(g wet sediment)$^{-1}$ in the 16-18 mm depth layer. The maximum 16S CB copies of *Ca.*
Electrothrix in station 2 corresponded to the maximum pH in depth. According to Gelhoed et
al. (2020) and using sediment density from the same stations obtained in 2019 (pers. comm.
M. Fouet), we calculated a CB density of 7.4 ± 0.4 and 74.4 ± 5.0 m.cm$^{-3}$ at stations 1 and 2
respectively.

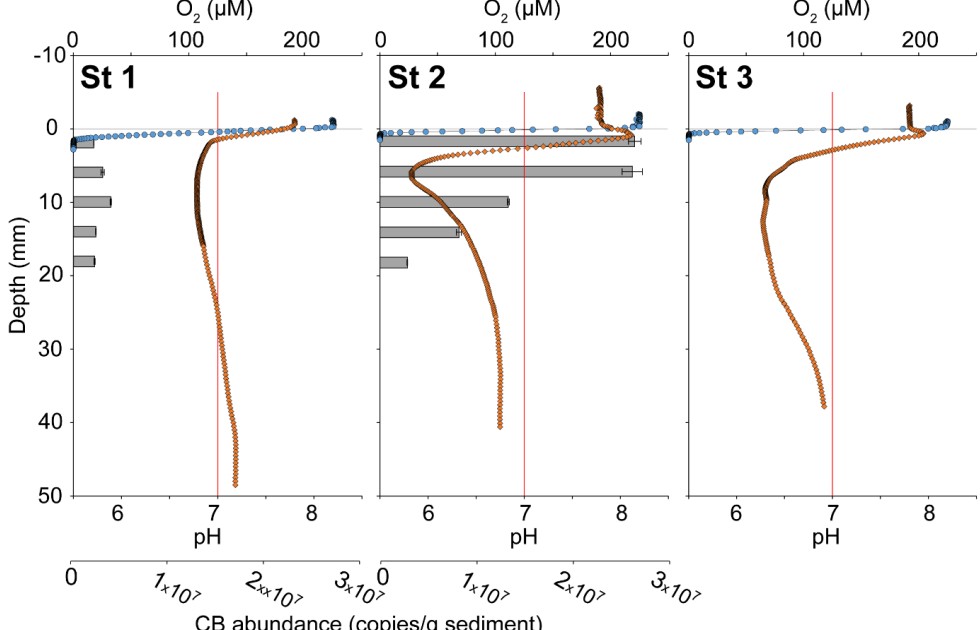

***Figure 4. Sediment oxygen (blue circles) and pH (orange diamonds) microprofiles at the three stations, and vertical distribution of cable bacteria abundance (qPCR of Ca. Electrothrix 16S rRNA gene copies, grey bars) for stations 1 and 2. 0 is the position of the Sea Water Interface (SWI). The vertical red line represents neutral pH. The oxygen profile presented is one of those obtained by microprofiling, and representative of O$_2$ penetration for each station.***

## 3.2 Hard-Shelled Benthic Foraminiferal
### 3.2.1 Living Foraminiferal Diversities and Densities
The foraminiferal species assemblages were typical of the estuarine environments (Debenay
et al., 2000), with a poor species richness (14, 13 and 18 species at stations 1, 2 and 3
respectively). *Ammonia* spp. and *Haynesina germanica* (Ehrenberg, 1840) both strongly
dominated the assemblages at all three stations (25.1 and 51.5 % respectively of the total
assemblage for station 1, 14.5 and 48.2 % for station 2, 7.3 and 61.4% for station 3; **Figure**
**5**). *Ammonia* spp. included the species *Ammonia veneta* (Schultze, 1854) (phylotype T1 after
Hayward et al., 2004), *Ammonia aberdoveyensis* Haynes, 1973 (phylotype T2 after Hayward
et al., 2004), and *Ammonia confertitesta* Zheng, 1978 (phylotype T6 after Hayward et al.,
2004). Agglutinated foraminifera represent 19.9, 25.7 and 12.7 % of the total assemblage at



stations 1, 2 and 3, respectively. They were dominated by *Ammobaculites agglutinans*
(d'Orbigny, 1846)*.*

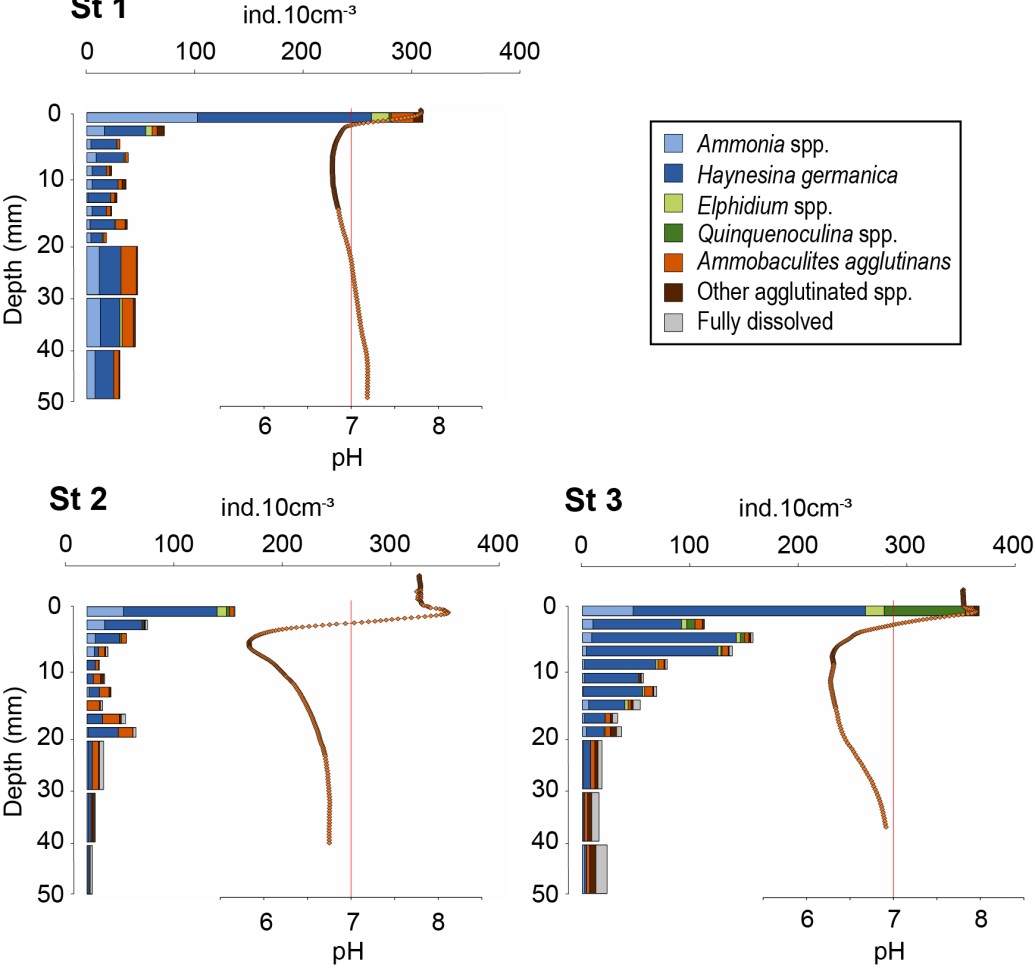

*Figure 5. Vertical distributions of living-foraminifera densities per 10 cm³ of sediment at the three stations (>125 µm fraction). 0 is the position of the Sea Water Interface (SWI). Recall of pH microprofiles (orange diamonds) and neutral pH (vertical red line).*

### 3.2.2   Living Foraminiferal Vertical Distribution

Total densities of CTG-labelled foraminifera in cores 1, 2 and 3 were 1273, 548, and 1431
ind.50cm⁻² respectively. Highest densities were found in the first layer of sediment (0-2 mm
depth) for all cores with 295, 137 and 371 ind.10cm⁻³ at stations 1, 2 and 3, respectively (**Figure**
**5**), where dioxygen was available and pH was maximal (**Figure 4**).
At station 1, total density dropped below 2 mm to stabilize at 30 ± 9 ind.10cm⁻³ (**Figure**
**5**). At station 2, the vertical distribution of total densities showed two maxima. The first at the
SWI and a second at 18-20 mm depth with 47 ind.10cm⁻³. A first minimum of 11 ind.10cm⁻³



was observed at 8-10mm depth close to the lowest pH layer and a second minimum of 5
ind.10cm$^{-3}$ was observed at the bottom of the core. At station 3, after a maximum at the SWI,
foraminifera density decreased gradually with depth, following the pH trend, to reach on
average 19 ± 4 ind.10cm$^{-3}$ from 20 to 50 mm depth.

At station 1, the ratio of calcareous foraminifera in the living foraminiferal assemblage

(C/T) was 0.91 for the SWI (**Table 3**) and around 0.77 ± 0.07 for the layers below. At station
2, C/T was 0.97 of the SWI and on average 0.64 ± 0.16 between 2- and 50-mm depth
(**Appendix**). However, agglutinated taxa dominated the assemblages from 10 to 18 mm, just
below the pH minimum, with a drop of C/T ratio to 0.39 ± 0.18 (**Appendix**). At station 3, the
C/T ratio was 0.97 at the SWI and decreased asymptotically as calcareous foraminiferal
densities vanished to reach 0.72 ± 0.15 below 20 mm after the pH minimum zone (**Appendix**).
**3.2.3   Calcareous Test Dissolution of Living Foraminifera**
**Figure 6** shows the dissolution stage (DS) of calcareous foraminifera for three selected layers
(0-2 / 6-8 / 40-50 mm) for living assemblages. At station 1, living specimens with calcareous
test showed low alteration. The DS remained stable through depth (p > 0.05). Specimens with
"Intact tests" (DS-0) or "slight surface dissolved tests" (DS-1) represented 90 % of calcareous
foraminifera. The strongest dissolution stages were DS-2 ("peeled test") and DS-3 ("cracked
test") accounting for less than 10 %.

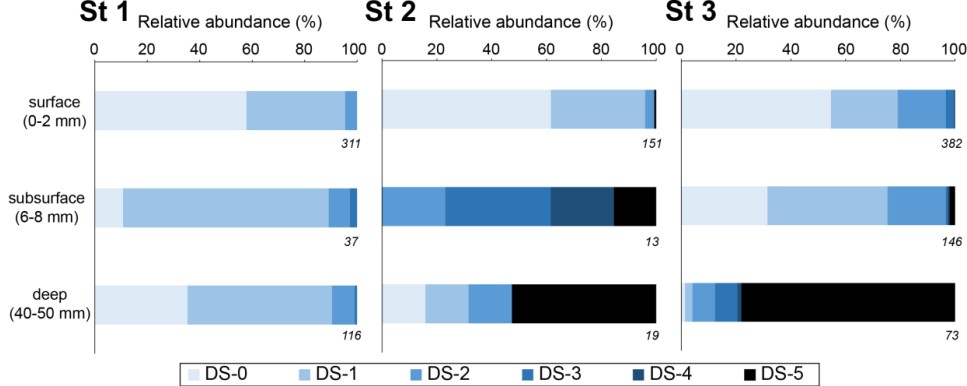

*Figure 6. Relative abundance of living benthic foraminifera with calcareous test for each dissolution stage for 10 cm$^3$ of sediment (Ammonia spp. and H. germanica from the >125 µm fraction). Three depth levels were analysed: the surface (0-2 mm; oxic zone), the subsurface (6-8 mm; suboxic zone corresponding to pH minimum), and the deeper (40-50 mm; anoxic zone). The numbers on the lower right of the boxes are the total numbers of SEM photographed specimens.*

Conversely, at station 2, many foraminifera were very fragile under manipulation.

Numerous "fully dissolved tests" (DS 5) with only the organic lining were observed through
depth (50 ind.50cm$^{-2}$; **Appendix**). At the SWI (0-2 mm layer), DS-0 and DS-1 tests represented
95% of the calcareous test foraminifera in the living assemblage. Only few DS-2 and DS-5
specimens were present. In the subsurface level (6-8-mm depth), corresponding to the most



acidic conditions, no DS-0 and DS-1 specimen were observed. DS-4 and DS-5 tests were
about 40 % of the calcareous tests observed. At the deepest layer (40-50 mm), DS-5
specimens were dominant (>50 %). The surface layer was significantly different (p < 0.005)
from the two deeper layers that showed no significant differences (p = 0.267).
At station 3, many foraminifera were fragile under manipulation, and DS-5 specimens
were abundant through depth with about 140 ind.50cm$^{-2}$ (**Appendix**). At the SWI (0-2 mm
layer) and in the subsurface level (6-8-mm depth), DS-0 and DS-1 specimens represented
about 75 % while DS-2 accounted for 20 %. Few specimens of DS-3, DS-4 and DS-5 were
observed. At the deepest layer (40-50 mm), DS-5 specimens were the most abundant
calcareous tests foraminifera (78 %). The severe DS (DS-3, 4 and 5) were significantly
overrepresented in the deep layer than in the surface and subsurface layers (p < 0.005). DS
were not significantly different between surface and subsurface (p = 1).
Overall, the exact Fisher's test revealed significant difference dissolution stages among
stations (p<0.005). The pair-wise Fisher's exact test showed that the low DS (0,1,2) were
significantly overrepresented at station 1 compared to the two other stations (p<0.005).
Furthermore, there were no significant difference between stations 2 and 3 (p=0.532).
**3.2.4  Calcareous vs. Agglutinated Foraminifera in the Dead Assemblages**
Species in the benthic foraminiferal thanatocoenosis were the same as in the living
assemblages. At station 1, calcareous taxa dominated agglutinated ones in the dead
assemblage with C/T ratio varying from 0.74 to 0.89 (**Table 3**). The proportion of organic lining
(DS-5/C) increased slightly with depth, from 0.06 to 0.18. On the other hand, at station 2,
agglutinated taxa dominated the dead assemblage in the surface and subsurface levels (C/T
ratio of 0.43 and 0.36 respectively) but not in the deepest one even if they remained abundant
(0.65; **Table 3**). The DS-5/C ratio was very high in all three depth layers, remaining >0.70. At
station 3, C/T ratio remained high in the dead assemblage of both surface and subsurface with
0.88, 0.83, and decreased strongly to 0.36 in depth where agglutinated specimens were
dominant. The DS-5/C ratio increased with depth, from 0.06 at the surface to 0.95 in the deeper
layer.
Comparing dead and living assemblages, it can be noted that for station 1, C/T ratio
were not very different whatever the depth (**Table 3**). Stations 2 and 3 showed much lower C/T
ratios in the dead assemblages indicating a marked loss of calcareous foraminifera during
taphonomic processes although this difference is not significant. In addition, stations 2 and 3
showed a higher occurrence of DS-5 tests in the dead assemblages resulting in high DS-5/C
ratios. In detail, station 2 showed the highest DS-5/C ratio in the subsurface layer (0.96) where
pH is minimal, while station 3 showed a strong increase of this ratio in the deepest layer (0.95).



**318  Table 3. Densities of living and dead foraminifera for each depth layer at the three all stations (ind.10 cm⁻³**
**319  of sediment).** *Depth correspondences: surface (0-2 mm), subsurface (6-8 mm) and deep (4-5 mm). "Calcareous"*
**320** *class includes the DS-5 specimens (fully dissolved test showing the organic lining). A = agglutinated, C =*
**321** *calcareous, ratios are those described in the methods.*

| | | Living foraminifera | | | | | Dead foraminifera | | | | |
|---|---|---|---|---|---|---|---|---|---|---|---|
| | | Agglutinated (A) | Calcareous (C) | Fully dissolved test (DS-5) | C/T ratio | DS-5/C ratio | Agglutinated (A) | Calcareous (C) | Fully dissolved test (DS-5) | C/T ratio | DS-5/C ratio |
| **St 1** | surface [0-2 mm] | 30 | 295 | 0 | 0.91 | 0.00 | 212 | 589 | 38 | 0.74 | 0.06 |
| | subsurface [6-8 mm] | 3 | 36 | 0 | 0.92 | 0.00 | 21 | 153 | 22 | 0.88 | 0.14 |
| | deep [40-50 mm] | 6 | 26 | 0 | 0.81 | 0.00 | 94 | 772 | 141 | 0.89 | 0.18 |
| **St 2** | surface [0-2 mm] | 4 | 137 | 1 | 0.97 | 0.01 | 373 | 282 | 197 | 0.43 | 0.70 |
| | subsurface [6-8 mm] | 7 | 12 | 2 | 0.63 | 0.17 | 181 | 104 | 100 | 0.36 | 0.96 |
| | deep [40-50 mm] | 1 | 4 | 2 | 0.80 | 0.50 | 239 | 453 | 327 | 0.65 | 0.72 |
| **St 3** | surface [0-2 mm] | 12 | 371 | 0 | 0.97 | 0.00 | 58 | 418 | 49 | 0.88 | 0.12 |
| | subsurface [6-8 mm] | 7 | 137 | 3 | 0.95 | 0.02 | 45 | 214 | 53 | 0.83 | 0.25 |
| | deep [40-50 mm] | 9 | 14 | 11 | 0.61 | 0.79 | 493 | 274 | 259 | 0.36 | 0.95 |

## 322  4 DISCUSSION

### 323  4.1  Cable Bacteria Density and Activity in Mudflats of the Auray
### 324  Estuary

Oxygen and pH microprofiles recorded at stations 2 and 3 showed the typical fingerprint of
CBA: a pH maximum within the oxic zone without oxygen production followed by a significant
acidification into the suboxic zone (**Figure 4**; Nielsen et al., 2010; Pfeffer et al., 2012; Risgaard-
Petersen et al., 2012; Meysman et al., 2015). The presence of CB at station 2 was further
confirmed by the qPCR data. At station 2, the 16S CB copy number was constant within the
oxic and suboxic zones (upper first centimetre; **Figure 4**). The calculated filament density of
about 70 m.cm⁻³ at this station was in the same order of magnitude than the *in situ* densities
reported from the Baltic sea (Marzocchi et al., 2018; Hermans et al., 2019), from bivalve reefs
(Malkin et al., 2017), subtidal mudflats (van de Velde et al., 2016) or intertidal salt marshes
(Larsen et al., 2015). The geochemical signature at station 1 is less clear regarding Cable
Bacteria Activity (CBA). There was no pH peak in the oxic zone and only a moderate
acidification within the suboxic zone ($\Delta$pH < 0.1). This acidification was lower than expected if
only driven by sulphate reduction and too high to be explained by iron reduction (van Cappellen
and Wang, 1996; Soetaert et al., 2007). Therefore, we suggested that it was driven by low
CBA. The qPCR data indicated very low CB filament density about 7 m.cm⁻³. The filament
density was in the low range of the *in situ* densities reported from the Baltic sea (Marzocchi et
al., 2018; Hermans et al., 2019). The diversity of pH microprofiles observed between the three



stations could indicate a contrasted intensity of the CBA between stations. According to the
low filament abundance and the low range of pH ($\Delta$pH = 1.0) at station 1, the CBA would be
minimal and it would have limited impact on the sediment geochemistry. Conversely, the strong
abundance and pH range ($\Delta$pH = 2.4) suggest the most intense CBA at station 2, whereas pH
range ($\Delta$pH = 1.8) at station 3 suggests an intermediate to high CBA.
Currently, the control factors of spatial and temporal variability of the CB density and
the CBA on mudflats are still unresolved. Our observations suggest that *Ulvae* mats observed
at stations 2 and 3 during core sampling could play a role on CB development. Several studies
showed that macrophyte decay is rather slow compared to mirophytobenthic mineralization
and favours free-sulphide production and upward diffusion (Anschutz et al., 2007; Metzger et
al., 2007; Cesbron et al., 2014; Delgard et al., 2016). Previous observations confirm the rather
high spatial and temporal CBA dynamics already mentioned in the literature (e.g. Seitaj et al.,
2015; Lipsewers et al., 2017; Hermans et al., 2019).

## 4.2 Impact of Cable Bacteria on Living Foraminifera

The CBA causes pH anomalies that impact sediment geochemistry and lead to the carbonate
dissolution process as described in Risgaard-Petersen et al. (2012), Meysmann et al. (2015),
van der Velde et al. (2016) and Rao et al. (2016). By analogy, it has been supposed that CBA
could also be responsible for foraminifera dissolution (Risgaard-Petersen et al., 2012; Richirt
et al., 2022). We showed in **Figure 4** and **Figure 6** that advanced dissolution stages 3, 4 and
5 were significantly overrepresented at stations 2 and 3, where acidification was important,
compared to station 1 where no DS-5 was observed. More precisely, vertical DS distribution
corresponded to vertical acidity variability at stations 2 (0.01 < DS-5/C < 0.50) and 3 (0.00 <
DS-5/C < 0.79). There is no indication for such depth distribution at station 1 where pH
variability was the lowest (DS-5/C = 0). The relative abundance of calcareous specimens over
agglutinated (C/T) is very stable along depth at station 1 (0.78 ± 0.07; **Appendix**) whereas this
ratio is more variable at stations 2 and 3 (0.65 ± 0.17 and 0.73 ± 0.15 respectively), confirming
that pH conditions affect assemblage composition through the under representation of
calcareous foraminifera although species diversity is never affected (**Figure 5**). Assuming that
acidification intensity in the suboxic zone is due to cable bacteria activity, our data suggest that
CB have a drastic effect on the integrity of shells from living benthic foraminifera and potentially
on their assemblages. The magnitude of this effect may depend in the CBA intensity and
duration throughout the life cycle of foraminifera.
Since the dataset of the present study is rather limited, one can examine literature data
that provides together oxygen and pH microprofiles with sub-centimetre vertical distribution of
living foraminifera in intertidal mudflats first and other benthic environments then. Geochemical





data from an intertidal mudflat of the Arcachon basin in the French Atlantic coast suggest CBA
in May 2011 at N station (Cesbron et al., 2016) with a ΔpH = 1.6 and a pH minimum of 6.2 well
below the oxic zone at 20-mm depth. At the same station in July 2011, all calcareous benthic
foraminifera specimens showed a fully dissolved test with the organic lining remaining (DS-5/C
= 1). The assemblage also showed that, *Eggerella scabra*, an agglutinated species, strongly
dominated the foraminiferal assemblage at all depths down to 50 mm, except for the 0 to 5
mm layer (C/T = 0.88 ± 0.02 for the uppermost layer; C/T = 0 below). The authors assumed
that test dissolution resulted from a strong acidification of the sediments due to an intense
remineralisation of the relict roots of *Zostera*. We can assume here that these roots provided
the refractory material that enhanced sulphate reduction (Anschutz et al., 2007; Metzger et al.,
2007; Cesbron et al., 2014; Delgard et al., 2016), providing enough free-sulphide to favour CB
development that drove the dissolution process as it probably happened at stations 2 and 3 of
the Auray estuary in the present study. However, Cesbron and co-workers also showed that
during winter (February 2011), foraminifera showed less dissolution due to a lower intensity of
diagenetic processes including free-sulphide production and probably benthic acidification.
These results underline the importance of the temporal variability of diagenetic processes that
influence pore water geochemistry including CBA (Seitaj et al., 2015; Lipsewers et al., 2017;
Hermans et al., 2019; Malkin et al., 2022), and eventually calcareous test integrity. It also
questions about time integration of pH conditions recorded in the foraminifera tests as
foraminifera may have mechanisms to buffer pH variations as suggested by different studies
(de Nooijer et al., 2009b, 2014; Toyofuku et al., 2017) or vertical migration strategies (Geslin
et al., 2004; Pucci et al., 2009; Koho et al., 2011; Hess et al., 2013). It could be assumed that
the dissolution of the calcareous foraminifera tests would respond to an integrated dynamics
over several months of exposure to the acidity caused by CBA, rather than immediately. These
dynamics should be investigated in the future in Auray estuary to better understand differences
of dissolution stages observed between stations. It can also be assumed that tolerance to
acidification may be species-dependent and needs detailed investigation.

Conversely, a tidal mudflat from another estuarine system of French Atlantic coast
seems not to show indices of CBA nor occurrence of dissolution on living foraminifera. Living
foraminifera form the Brillantes mudflat of Loire estuary was studied at two stations in
September 2012 and April 2013 (Thibault de Chanvalon et al., 2015, 2022). The vertical
distribution of living foraminifera reported in the Loire mudflat was similar to the vertical
distribution of station 2 reported in the present study with a maximal density at the topmost
layer within the oxic zone, a minimal density around 10-mm depth and a second maximum
below. However, no foraminiferal test dissolution was reported by Thibault de Chanvalon and
co-workers and the foraminiferal assemblages were heavily dominated by calcareous



foraminiferal species, resulting in a DS-5/C ratio equal to zero and a C/T ratio about 1.
Furthermore, at these stations, pH profiles did not show signs of CBA at different occasions
(May 2013, February 2014, June 2018, unpublished data). pH decrease corresponded to
oxygen uptake and was below 0.5 units with a minimum about 7.7. No pH peak at the interface
was observed in a profile performed in the dark neither. The major difference between these
systems is the size of the river that induces significant resuspension-deposition event for the
Loire estuary limiting the development of favourable conditions to CB development. In addition,
bioturbation seems to be intense at the Brillantes mudflat (Thibault de Chanvalon et al., 2015,
2016b, 2017). Another difference between these studies is the absence of macrophytes at the
studied stations of the Brillantes mudflat. Finally the size of the catchment area of Loire
provides an important flux of suspended particles rich in metallic oxides that will once settled
in the mudflat generate a thick layer of sediment where the iron cycle dominates diagenetic
processes acting as an efficient "iron curtain" that maintains free-sulphide between 5 to 10 cm
depth (Thibault de Chanvalon et al., 2016b, 2016a) to be out of reach for CB. These combined
conditions are not favourable to CB development (Malkin et al., 2014, 2017). This, foraminiferal
observations strongly suggest the absence of CBA in the studied part of the Brillantes mudflat.
This area may be considered as a control station.

Other studies reporting calcareous test dissolution of benthic living foraminifera in
comparable environments are published but without geochemical data, allowing to discuss
potential CBA (Alve and Nagy, 1986; Buzas-Stephens, 2005; Polovodova and Schonfeld,
2008; Bentov et al., 2009; de Nooijer et al., 2009; Kurtarkar et al., 2011; Haynert et al., 2012;
Schönfeld and Mendes, 2022). Although the hypotheses put forward by these authors on the
causes of test dissolution are all plausible (environmental pollution, freshwater inputs, organic
matter degradation), they are not strongly explained. Therefore, the absence of pH data
(Buzas-Stephens, 2005; Polovodova and Schonfeld, 2008) or its insufficient vertical resolution
(Alve and Nagy, 1986; Haynert et al., 2012; Schönfeld and Mendes, 2022) does not exclude
the potential involvement of CB in those environments. In the Baltic sea, that could be
considered as a sort of giant estuary, Charrieau et al. (2018a), provide pH microprofiles that
seem to indicate the absence of CBA. However, these authors observed calcareous test
dissolution of living foraminifera and concluded that dissolution may be the consequence of a
complex set of environmental factors whose ecological equilibrium can change rapidly in such
coastal areas (salinity, oxygen concentration, pH and $\Omega_{calc}$). Laboratory experiments conducted
by these authors (Charrieau et al., 2018c), seem to indicate that low salinity may be an
important factor on calcareous test dissolution. The difference with estuarine studies discussed
above is probably that salinity change dynamics in the Baltic is rather minor compared to





salinity in Auray and Loire that are macrotidal systems with species adapted to such salinity
variations.
Considering the increase of observations of cable bacteria activity occurrence in a wide
range of marine environments (Malkin et al., 2014; Burdof et al., 2017) like estuaries, coastal
lagoons, salt marshes, marine lakes, tidal and subtidal mudflats, we assume that the impact
of CBA on foraminifera community should be strongly considered.

## 4.3   Impact of Cable Bacteria on Dead Assemblages

Richirt et al. (2022) have assumed that calcareous test dissolution resulting from CBA could
be responsible for low densities of calcareous tests in the dead assemblages recorded in
sediments of Lake Grevelingen. Our results suggest that acidification caused by CBA strongly
affects the calcareous shell integrity and the assemblage composition of living foraminifera
before taphonomic processes. Our study also suggests that after foraminifera death, CBA
keeps transforming the foraminifera assemblage during test burial confirming the hypothesis
formulated by Richirt and coworkers (2022). Comparing C/T and DS-5/C ratios between living
and dead assemblages at different depths we relate in detail the impact of pH distribution and
therefore CBA to the taphonomic loss. Under a weak CBA (like at station 1), calcareous tests
were relatively well preserved. At this station, the community structure between living and dead
assemblages varied slightly (C/T ranged from 0.81- to 0.98 in living assemblage and from 0.74-
to 0.89 in dead assemblage). The occurrence of dissolution in the living assemblage was nil
while in the dead assemblage the DS-5/C ratio increased with depth from 0.06 to 0.18
indicating that the low dissolution generated a relatively slow taphonomic process. Calcareous
tests dominated both living and dead assemblages with an increase of this trend with depth in
the dead assemblage confirming the good preservation of calcareous foraminifera. Where
CBA was moderate (like at station 3), the dissolution effect on the thanatocoenosis was gradual
with depth. Calcareous test density decreased through the wide acidic layer (C/T decrease
from above 0.8 to 0.36 at 50-mm depth) and there was an accumulation of fully dissolved tests
showing only their organic linings in dead foraminiferal assemblages at depth (DS-5/C of 0.95).
This feature suggests that the moderate dissolution generated a gradual taphonomic process
leading to a noticeable calcareous loss with depth. Eventually, under a strong and intense CBA
(like at station 2), the dissolution effect occurred mostly within the restricted acidic layer. The
calcareous tests disappeared from the dead foraminiferal assemblage in this subsurface layer
while the fully dissolved tests showing only their organic linings and agglutinated tests
accumulated (C/T = 0.37 and DS-5/C = 0.96). At depth, the dead foraminiferal assemblage
showed fairly high densities that are comparable to stations where CBA was less intense. As
the living specimens were quite rare, such accumulation of dead tests suggested that
somehow they bypassed the acidic firewall of the suboxic layer. If tests arrived at depth through



sedimentary burial the acidic firewall was possibly not active in a recent past. If CBA is not
recent, physical or biological reworking buried sufficiently fast to preserve tests from corrosive
conditions and mechanic crumbling. At this stage, these hypotheses cannot be assessed. One
can note the high concentration of dead fully dissolved tests in the first 2 mm (0.70) where pH
is the most alkaline suggesting that sedimentary reworking may have brought dead specimens
from the subsurface acidic layer to the surface. Further studies on dead assemblages are
needed to statistically validate the CBA vs. calcareous test loss relationship.
With low pH in pore water, the dissolution process resulting from CBA suggests an
imprint on taphonomy and on historical records yet to be explored. Dissolution of living and
dead calcareous test foraminifera due to CBA may be taken into account in interpretations of
foraminiferal assemblages in historical studies. As proposed in Richirt et al. (2022), historical
records of benthic foraminifera could also be used to reconstruct past CBA and determine the
age of the first CB occurrence in the studied environments. Therefore, associating it with major
environmental changes, light could be shed on the original factors of this bacterial spreading
discovered only ten years ago.

## 5 CONCLUSION

This original study strongly suggests that sediment acidification caused by CBA could be
responsible for significant calcareous test foraminifera dissolution patterns. As a result,
proportions of calcareous test would change in both living and dead assemblages. The
proportion of fully dissolved tests showing only their organic linings would increase in the living
assemblages in the suboxic and anoxic zones of the sediment, as well as in the
thanatocoenosis. The spatial dynamics of calcareous test dissolution in mudflats described in
the present study seems to be the consequence of CBA which leads to a wide range of pore
water pH in the suboxic zone of sediment.
Now that we have an idea of the potential impact of this bacterial activity on
foraminiferal assemblages and on calcareous test preservation, we are entitled to ask what
implications this might have for environmental interpretations of data from their use as
paleoproxies, or bioindicators. In order to better understand this impact, it would be relevant to
explore *in situ* and *in vitro* the effect of CBA at several time scales on the resilience of
foraminiferal communities to learn more about the integration of their response in the
historic/fossil record. Eventually, we should be able to provide a historical retrospective on the
presence of CB in marine sediments and their impact on the benthic meiofauna. In this
perspective, we should combine such studies with the development of biomarkers of these
chemolithoautotrophic bacteria or of ancient eDNA.



*Data availability.* All of the data are published within this paper and in the Supplement. The
raw data used to make the figures are available on request.
*Author contributions.* Maxime DAVIRAY (foraminiferal and geochemical analysis, writing,
review and editing), Emmanuelle GESLIN (head of CB-For CNRS project, foraminiferal
analysis, writing, review and editing), Nils RISGAARD-PETERSEN (statistical inference, writing,
review and editing), Vincent SCHOLZ (qPCR proceeding, review and editing), Marie FOUET
(field work, review and editing), Edouard METZGER (microprofile acquisition, writing, review and
editing).
*Competing interests.* At least one of the (co-)authors is a member of the editorial board of
Biogeosciences.
*Acknowledgments.* The authors are grateful to Sophie QUINCHARD for assistance with the
foraminifera picking, Sophie SANCHEZ for sample splitting (LPG, Université d'Angers) and
Romain MALLET (SCIAM, Université d'Angers) for the achievement of a part of the SEM
imaging. Susanne NIELSEN and Ian MARSHALL (Aarhus University) are thanked for assistance
with the qPCR analysis. The authors are grateful to Frans JORISSEN for field work, and his
advice for the writing process (LPG, Université d'Angers). The authors thank the participants
to the field trips (2019 and 2020).
*Financial support.* The authors received funding from the CNRS-INSU (program LEFE-
CYBER, project CB-FOR), from the Angers University and from the OFB (project FORESTAT).

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

***Appendix.*** *Foraminiferal absolute densities and ratios in the Auray estuary for the three stations.*

| | Depth layer | Layer volume (cm-3) | *Haynesi na ger mani ca* | *Am mon ia* spp. | Elph idiu m spp. | *Quinqu eloculin a* spp. | DS-5 spe cim en | *Ammob aculites aggluti nans* | Other agglu tinan s | Tot al | C / T rati o | DS-5 / C ra ti o |
|---|---|---|---|---|---|---|---|---|---|---|---|---|
| St 1 | [0-2 mm] | 10.6 | 179 | 113 | 18 | 2 | 0 | 23 | 9 | 344 | 0.91 | 0.00 |
| | [2-4 mm] | 10.6 | 42 | 18 | 6 | 0 | 0 | 5 | 7 | 78 | 0.85 | 0.00 |




| | | | | | | | | | | | |
|---|---|---|---|---|---|---|---|---|---|---|---|
| [4-6 mm] | 10.6 | 26 | 4 | 0 | 0 | 0 | 3 | 0 | 33 | 0.91 | 0.00 |
| [6-8 mm] | 10.6 | 28 | 9 | 1 | 0 | 0 | 3 | 0 | 41 | 0.93 | 0.00 |
| [8-10 mm] | 10.6 | 15 | 5 | 0 | 0 | 0 | 3 | 2 | 25 | 0.80 | 0.00 |
| [10-12 mm] | 10.6 | 26 | 5 | 0 | 0 | 0 | 4 | 4 | 39 | 0.79 | 0.00 |
| [12-14 mm] | 10.6 | 23 | 1 | 0 | 0 | 0 | 4 | 2 | 30 | 0.80 | 0.00 |
| [14-16 mm] | 10.6 | 15 | 5 | 0 | 0 | 0 | 4 | 1 | 25 | 0.80 | 0.00 |
| [16-18 mm] | 10.6 | 25 | 3 | 1 | 0 | 0 | 9 | 2 | 40 | 0.73 | 0.00 |
| [18-20 mm] | 10.6 | 12 | 4 | 1 | 0 | 0 | 3 | 0 | 20 | 0.85 | 0.00 |
| [20-30 mm] | 52.8 | 110 | 61 | 2 | 0 | 0 | 77 | 6 | 256 | 0.68 | 0.00 |
| [30-40 mm] | 52.8 | 99 | 67 | 11 | 0 | 2 | 57 | 9 | 245 | 0.73 | 0.01 |
| [40-50 mm] | 52.8 | 93 | 42 | 2 | 0 | 0 | 28 | 4 | 169 | 0.81 | 0.00 |
| [0-2 mm] | 10.6 | 95 | 37 | 9 | 3 | 1 | 4 | 0 | 149 | 0.97 | 0.01 |
| [2-4 mm] | 10.6 | 38 | 18 | 1 | 0 | 2 | 1 | 1 | 61 | 0.97 | 0.03 |
| [4-6 mm] | 10.6 | 24 | 8 | 2 | 0 | 0 | 5 | 0 | 39 | 0.87 | 0.00 |
| [6-8 mm] | 10.6 | 4 | 7 | 0 | 0 | 2 | 6 | 1 | 20 | 0.65 | 0.15 |
| [8-10 mm] | 10.6 | 8 | 0 | 0 | 0 | 1 | 3 | 0 | 12 | 0.75 | 0.11 |
| [10-12 mm] | 10.6 | 6 | 0 | 0 | 0 | 1 | 7 | 3 | 17 | 0.41 | 0.14 |
| [12-14 mm] | 10.6 | 11 | 2 | 0 | 0 | 0 | 10 | 2 | 25 | 0.52 | 0.00 |
| St2 [14-16 mm] | 10.6 | 0 | 0 | 0 | 0 | 2 | 13 | 1 | 16 | 0.13 | 1.00 |
| [16-18 mm] | 10.6 | 16 | 0 | 0 | 0 | 4 | 17 | 2 | 39 | 0.51 | 0.20 |
| [18-20 mm] | 10.6 | 31 | 1 | 0 | 0 | 3 | 15 | 0 | 50 | 0.70 | 0.09 |
| [20-30 mm] | 52.8 | 22 | 6 | 2 | 0 | 20 | 33 | 6 | 89 | 0.56 | 0.40 |
| [30-40 mm] | 52.8 | 15 | 5 | 0 | 0 | 4 | 5 | 8 | 37 | 0.65 | 0.17 |
| [40-50 mm] | 52.8 | 9 | 0 | 0 | 0 | 10 | 5 | 1 | 25 | 0.76 | 0.53 |





| | | | | | | | | | | | |
|---|---|---|---|---|---|---|---|---|---|---|---|
| | [0-2 mm] | 10.6 | 238 | 52 | 19 | 83 | 0 | 8 | 5 | 405 | 0.97 | 0.00 |
| | [2-4 mm] | 10.6 | 91 | 11 | 5 | 8 | 0 | 7 | 2 | 124 | 0.93 | 0.00 |
| | [4-6 mm] | 10.6 | 148 | 9 | 4 | 4 | 2 | 4 | 2 | 173 | 0.97 | 0.01 |
| | [6-8 mm] | 10.6 | 133 | 4 | 3 | 1 | 3 | 6 | 1 | 151 | 0.95 | 0.02 |
| | [8-10 mm] | 10.6 | 73 | 2 | 2 | 0 | 2 | 6 | 1 | 86 | 0.92 | 0.03 |
| | [10-12 mm] | 10.6 | 55 | 2 | 1 | 0 | 2 | 1 | 1 | 62 | 0.97 | 0.03 |
| | [12-14 mm] | 10.6 | 60 | 1 | 2 | 0 | 3 | 8 | 1 | 75 | 0.88 | 0.05 |
| St3 | [14-16 mm] | 10.6 | 37 | 6 | 3 | 0 | 7 | 3 | 2 | 58 | 0.91 | 0.13 |
| | [16-18 mm] | 10.6 | 21 | 2 | 0 | 0 | 5 | 5 | 2 | 35 | 0.80 | 0.18 |
| | [18-20 mm] | 10.6 | 18 | 4 | 1 | 0 | 5 | 5 | 6 | 39 | 0.72 | 0.18 |
| | [20-30 mm] | 52.8 | 38 | 4 | 0 | 0 | 21 | 20 | 16 | 99 | 0.64 | 0.33 |
| | [30-40 mm] | 52.8 | 7 | 4 | 1 | 0 | 35 | 15 | 19 | 81 | 0.58 | 0.74 |
| | [40-50 mm] | 52.8 | 8 | 9 | 1 | 0 | 57 | 15 | 32 | 122 | 0.61 | 0.76 |
