# Peer review of "POTENTIAL IMPACTS OF CABLE BACTERIA ACTIVITY ON HARD-1 SHELLED BENTHIC FORAMINIFERA: IMPLICATIONS FOR THEIR 2 INTERPRETATION AS BIOINDICATORS OR PALEOPROXIES 3"

_Biogeosciences, 2023_

## Author Comment (AC1)

**Response to Reviewer 1: Sebastiaan van de Velde**

(https://doi.org/10.5194/bg-2023-169-RC1)

**General comments:**

In this manuscript, Daviray et al. evaluate the impact of porewater acidification by the activity of cable bacteria on the preservation of benthic foraminifera in coastal marine sediments. The manuscript is well structured and the methods and aims were clear. My main comment would be that the introduction and discussion are not very in depth, and focus on one singled out factor (cable bacteria), while having not providing more context or considering other environmental factors. I have listed a few specific comments/questions below that could improve the impact of the manuscript. Overall, I believe this is a solid research paper, but some of the discussion should be a little more in-depth.

**Specific comments:**

The authors state that the aim of the study is to investigate the impact of cable bacteria activity on the preservation of forams. Throughout the MS they focus almost exclusively on this aim, and by doing so they provide very little broader context of why this work is important. I was left with a few questions that could be addressed by restructuring the introduction and discussion.

- Why is understanding the environmental parameters that control foraminifera important? As presented in the introduction (l. 58-62), to refine benthic foraminifera as bioindication tools or palaeoproxies, environmental parameters that influence species distribution, population dynamics, shell chemical composition and structuring... have to be better understand. This study is part of this effort to identify the causal factors of geochemical changes in microhabitats that have an impact on foraminifera.

- Are there any other processes that cause porewater acidification comparable to cable bacteria (e.g. reoxidation of reduced species in the oxic zone could lead to an acidic minimum), and why are they not considered? We agree to the reviewer and discuss about such processes in Discussion lines 336-338. As the sediment acidification continued well below the oxic zone, oxic processes did not appear to be involved here in such pH decrease. However, acidification due the reoxidation of reduced iron by nitrate or $MnO_2$ in the suboxic zone could be more discussed (Soetaert et al., 2007; Middelburg et al., 2020). First, important remobilization of iron can be seen without test dissolution within the suboxic zone (Thibault de Chanvalon et al, 2015). Then, in Auray mudflats, nitrate is not a major component of bottom water and porewater chemistry in autumn as we have seen in other studies (< 4 µM, pers. comm.) so this hypothesis could be ruled out. Eventually, the involvement of $MnO_2$ can be discussed here, despite the lack of data. CBA involves iron reoxidation by $MnO_2$ within the suboxic zone (Sulu-Gambari et al., 2016). Then, this reaction seems to be a consequence of this bacterial activity, as is the sediment acidification that it may induce.

- Are there other locations aside from estuaries where these findings could be important? All marine environments where CB and carbonate-shelled benthic meiofauna may cohabit would be concerned (e.g. salt marshes, mudflats, lagoons, reefs, marine lakes; Burdof et al, 2017). Has this any impact on, e.g., climate records based on foraminifera isotopes? As things stand, there is no answer to this question. We can only assume that this bacterial activity could influence the isotopic composition of the foraminiferal test, if not a total loss of calcareous species in the sediment records, as Richirt and coauthors hypothesised (2022). This is why we open in the conclusion (l. 510-511) that

foraminifera could be used as paleoproxies of this bacterial dissolution process. Further investigations on this way should be performed.

Another factor that does not receive much consideration is the seasonality. So far, all laboratory studies of cable bacteria have shown a boom-and-bust cycle (rapid growth, followed by a collapse). Field studies on the other hand show a distinct seasonal pattern of alternations between cable bacteria, bioturbated macrofauna and Beggiatoa (e.g. Seitaj et al., 2015) or between cable bacteria and other sediment disturbance reworking events (e.g. van de Velde 2018). Indeed, the boom-and-bust cycle of CB in laboratory studies are observed, and the seasonal alternation of the sulphur-oxidising bacteria community on the field according to the hypoxia events inducing pH seasonal variability (Seitaj et al., 2015; Lipsewers et al., 2017; Malkin et al., 2022). However, such desoxygenation or reworking events have not been reported in this study area (Marie Fouet thesis; OFB and IFREMER data). Furthermore, the intertidal mudflats are reoxygenated at each low tide which could lead to the reactivation of cable bacteria activity in highly eutrophic environments. Unfortunately, there is little literature on cable bacteria activity under tidal cycle.

We agree that carrying out additional campaigns through time and laboratory experiments represent important issues for the future of the project. The seasonal alternation of bacterial communities is not the subject of this study and to enter these considerations risk weighing down the discussion. Discussing the potential temporal variability of CB and time integration by foraminifera as perspective seems sufficient to us for now.

So far, there has been no study that showed a constant presence of cable bacteria throughout a year, so it is likely that your site also experiences reworking by fauna or other resuspension events. You partly allude to the importance of sediment mixing at L483, but this is not considered anywhere else in the MS. What is benthic fauna community at the field site? We agree with the critic. This prospect of sediment mixing was mentioned without giving a possible cause, as was the macrofauna. The benthic macrofauna (> 2 mm) of the mudflat is dominated by polychaetes (*Nephtys* spp.) known to burrow into the sediments (Michaud et al, 2021). There are also bivalves (*Cerastoderma edule*), as well as a few gastropods (*Peringia ulvae*) and arthropods (*Chaetogammarus marinus* and *Apohyale prevostii*). Total abundance is around 15 ind.50 cm$^{-2}$ (pers. comm. Oihana Latchere).

Are there many intense resuspension events (e.g. storm floods)? The field survey was carried out at the end of September, at the end of the low-water period. There were no floods or storms in the weeks preceding our sampling (archives Météo France). The Auray estuary and the Morbihan gulf are very enclosed systems mostly protected from marine storms. The most intense resuspension phenomenon here would be rising tide (Menier and Dubois, 2011; Menier et al, 2011) and bioturbation.

How would they influence your interpretation (e.g. seasonal sediment mixing homogenizes the top sedimentary layers and moves forams from within the acidic zone down to the deeper layers, or vice versa; see, e.g. Hülse et al., 2022)? We can assume that a homogenization phenomenon in the upper sediment layers, under biotic or abiotic influence, would reduce CB activity as observed on mudflats (Malkin et al, 2014, 2022; Aller et al, 2019). We would then expect an increase in pH in the suboxic zone, a weakening of the dissolution process, and a shorter time residence within the acidic zone. The calcareous shells of the foraminifera would then be less likely to be subject to decalcification and would probably be better preserved. Levels of decalcification would be lower (< stage 3), and calcareous shell specimens would probably remain in the majority of assemblages.

Eventually, if calcareous foraminifera are decalcified so intensely, this means that despite the strong physical and biogeochemical dynamics of this kind of transitional environment in time and space, the corrosive conditions are sufficiently strong in intensity through time to generate dissolution in living organisms that are able to fight off these hostile conditions to a greater or lesser extent.

Finally, I am not entirely convinced you can use forams to reconstructing the history of cable bacteria, there are so many parameters that cause dissolution that it will be impossible to relate this robustly to cable bacteria activity (let alone reworking of the sediment - see Hülse et al., 2022). How would you go about doing that? This would involve a multivariate approach coupling (1) the identification of lipid biomarkers in cable bacteria or eDNA and their investigation in ancient sediments to determine their presence, (2) the study of foraminiferal species assemblages (C/T ratio), shell preservation and isotopic shell composition and (3) paleoenvironmental methods (like sedimentology) distinguish other factors responsible for the dissolution process and to infer it to bacterial activity.

Technical comments:

Title: 'a prelude' -> why not just say 'implications for …'? It can be; it was just a personal fantasy...

L32: strongly -> omit. At several instances you use 'strongly', e.g. 'strongly consider', 'strongly explained'. In get this is for emphasizing the importance, but you can omit strongly on most of these occasions. The importance of your results is clear for scientists working in your field. This suggestion will be taken into account.

L115: why only station 1 and 2? The data from station 3 were added to the manuscript after the DNA analyses had been carried out on the samples from stations 1 and 2; the apparatus and the team of the Microbiology Institute of Biology in Aarhus University (Denmark) were subsequently no longer available to us to carry out measurements for the three stations together.

L123: Is this tip diameter or length of your tip? We have specified "tip diameter" in the manuscript.

L125: for microsensor profiling, you should not have a stepsize smaller than the tip size? We are aware of this methodological limitation and the theory about microsensors. However, Unisense is not able to make pH robust probes thinner than 500 µm for *in situ* investigations. We tried many times and most of them did not last until the end of the first profile…

L235: Geelhoed The correction will be done.

L337: pH minima are also generated by reoxidation of reduced iron species (and other reduced species) We agree. For us, this was the meaning of "iron reduction". As mentioned above, we're going to discuss these reactions in a little more detail.

L376: remove then This suggestion will be taken into account.

L507: But it is specifically the low pH generated by cables that is important? Very good point. We agree that the *in situ* variability of the dynamics of CB activity from one mudflat to another under the same hydrological system is a very interesting prospect that deserves to be studied further such as their temporal dynamics.

References:

Seitaj et al., 2015: cited in MS

van de Velde et al., 2018: 10.1038/s41598-018-23925-y

Hülse et al., 2022: 10.1016/j.earscirev.2022.104213

Soetaert et al., 2007 (10.1016/j.marchem.2007.06.008)

Middelburg et al., 2020 (10.1029/2019RG000681)

Thibault de Chanvalon et al, 2015 (10.5194/bg-12-6219-2015)

Sulu-Gambari et al., 2016 (10.1016/j.gca.2016.07.028)

Burdof et al, 2017 (10.5194/bg-14-683-2017)

Richirt et al, 2022 (10.1016/j.palaeo.2022.111057)

Lipsewers et al., 2017 (10.1128/AEM.03517-16)

Malkin et al., 2022 (10.1002/lno.12087)

Marie Fouet's thesis

OFB (https://professionnels.ofb.fr/fr/node/276)

IFREMER (https://wwz.ifremer.fr/envlit/DCE/La-DCE-par-613 bassin/Bassin-Loire-Bretagne)

Michaud et al, 2021 (10.1357/002224021834670801)

Météo France
(https://donneespubliques.meteofrance.fr/donnees_libres/bulletins/BCMR/BCMR_03_202309
.pdf)

Menier and Dubois, 2011 (Carte 7137G Natures de fond du Golfe du Morbihan à 1/20 000,
SHOM Ed.)

Menier et al, 2011 (https://observatoire-littoral-morbihan.fr/wp-
content/uploads/2018/06/Menier-et-al.-2011.pdf)

Malkin et al, 2014 (10.1038/ismej.2014.41)

Aller et al, 2019 (10.1126/sciadv.aaw3651)

---

## Author Comment (AC2)

Response to Reviewer 3 (https://doi.org/10.5194/bg-2023-169-RC3)

General comments:

This manuscript examines the ecological and environmental impact of the recently investigated cable bacteria on the geobiology of sediments in brackish environments, and their effect on the preservation of tests of calcareous foraminifera inhabiting these environments. Overall, the methods, results, and discussion demonstrate a scientific rigor that should engage the Biogeosciences readership. However, there's a need for further exploration from the foraminifera perspective, which this study initiates.

The primary concerns of this reviewer include the absence of qPCR results at Station 3, and a perceived lack of adequate consideration regarding the carbonate saturation state in discussing carbonate test dissolution, which is deemed a significant question. As explain to Reviewer 1, we encountered some administrative limitations during our collaboration with the Microbiology Institute of Biology in Aarhus University (Denmark). In this regard, we would like to make it clear in the Method section by adding:: "For administrative reasons, it was only possible to carry out these DNA analyses for stations 1 and 2."

About the carbonate saturation state: a second parameter of the carbonate system is needed in addition to pH to calculate it (such as DIC or alkalinity). We agree that this information is relevant to our subject but we have no such data, which is a shame. However, the stage of dissolution of the calcareous shells seems sufficiently edifying here to support the discussion about dissolution process.

Furthermore, detailing the ecological aspects of each station, such as the visual characteristics of the sediments, grain size distributions, limnological settings etc., could enhance the discussion on the study's results, yet such information is notably absent, for instance, in Table 1. It is recommended that these issues be addressed either through revision or in subsequent publications within Biogeosciences. Majority of those ecological parameters have been studied and have been discussed in Fouet (PhD report, 2022). She discusses the granulo-hydodyn-marine influence link and its correlation with the diversity and relative abundance of some foraminiferal taxa in the Auray estuary that are more closely associated with the marine environment. A cross-reference to this work can be added to Table 1.

The comments are noted in below:

P1L7 Change bold text to normal. This is the affiliation required by the laboratory and the university with which we are associated.

P3L72 In this context, Charrieau et al. (2018c) and Charrieau et al., 2022 https://doi.org/10.1038/s41598-022-10375-w to mention may also contribute to the depth of the discussion. We agree that Charrieau et al (2018c) contributes to the specific point about test dissolution below a certain pH value for estuarine foraminiferal species (*Ammonia* sp. and *Elphidium* sp.). Charrieau et al (2022) focuses on a large symbiont-bearing benthic species in warm coral reef environments (*Peneroplis* spp.), which is very different from our case study. This is the reason why we did not include it.

P5L113 The authors stated that the samples were washed with tap water, but it is unclear whether the weaken/broken tests were affected by the flow of water. Can authors confirm if any evaluation was made on this matter? Also, is it possible to observe the same tests in their

natural setting by sorting raw samples and immersing them in seawater? This question arises from the fact that seawater usually has a carbonate saturation level greater than 1, which does not promote further dissolution. We agree that washing with seawater would probably be more appropriate, but we do not have such a system at Angers due to our distance from the coast. However, the Loire river providing tap water is relatively hard as it crosses the calcareous Parisian basin. The washing method used during this study was the same as for other projects on other marine sites carried out in the laboratory at the same time. No signs of dissolution of the foraminiferal tests were specifically reported that could incriminate this washing method. It will be pointed out in the Methods section that, to avoid further damaging the already fragile shells or organic linings, the wash was quick and gentle.

P6L131 Do authors measure carbonate saturation state or other related factors (e.g., calcium concentration or alkalinity) to understand calcium carbonate dissolution? As previously stated, we unfortunately do not have any chemical data other than pH and $O_2$ for further study of the saturation state of carbonates.

P6L147 I fully support the decision made by the authors. In my opinion, it is necessary to address the issue at the genus level and there is no need to narrow it down to the species level. Nonetheless, if there are any references that demonstrate species compositions that are unique to studied water, those should be included. We agree with the reviewer. We are not aware of any such studies to date.

P6L149 Authors use "living" to refer to fixed individuals stained with CTG, not live sorting. Absolutely, as mentioned in the Methods section.

P6L156 Charrieau et al. 2022, who conducted an experiment on *Peneroplis*, should be also introduced here. We agree and we will add it.

P10L176 I acknowledge that it is important to perform this calculation, but I need an explanation as to why the low ratio of calcareous foraminifera is regarded as a "loss" by the authors. Given that our assemblages are dominated by specimens with a calcareous test, and that previous studies on this site state the same (Redois et al, 1998; Fouet et al, 2022), we consider that a low ratio of calcareous specimens between the living and dead assemblages corresponds to a "loss" of calcareous foraminifera.

P10L199 It appears to be a reasonable statistical process. We thank the reviewer.

P11L217 Please specify why bacteria are not quantified in St. 3. Also, specify why quantification is done at St. 1 and 2 but not at St. 3. We will add a sentence about it: "For administrative reasons, it was only possible to carry out these DNA analyses for stations 1 and 2."

P11L229 Should pointing out or certifying if this distribution is a fingerprint be moved to the "Discussion" section? And, please provide citations or evidence that this distribution is "typical". We agree the reviewer and will delete this sentence from the "Results" part because of the repetition lines 326-327 in the "Discussion" part.

P12L238 Is the CB density zero at St. 3? Or is it missing? As we have said previously, we will precise that we do not have this data for St 3.

P14L278 If the experimenters took care to use tap water and a gentle water flow while washing, it is important to include a thorough description of this in the methodology section. This information is crucial for others to be able to replicate the experiment accurately. We agree and we will precise it in "2.2 Sediment Sampling and Processing" part (l. 113).

P16L338 It is currently known that the presence of pore water in sediments is determined by sulfate, iron reduction, and CBA. However, it is still unclear to readers whether any deposits of calcium carbonate organisms, other than foraminifera, exist in the area. For example, there may be shells of bivalves such as clams, or sea snails, which shells are rich in calcium carbonate. These calcium carbonate contents can buffer the pH, but there is no information available on sediment composition, including alkalinity or calcium carbonate content. Hence, it is important to explain why CBA can be attributed to this. Is this a logical conclusion based on previous studies? We have observed few bivalve and snail shells within the sampled sediment. A study has been carried out on CBA in muddy bivalve reefs (Malkin et al, 2017). They reported no dissolution of living shellfish. They concluded that $CaCO_3$ and alkalinity accumulated on the reef were remobilised by the CBA from the sediments towards the bottom waters. The $CaCO_3$ dissolution process was therefore very active. It is likely that the dissolution process plays a different role depending on the scale considered. Indeed, the surface/volume ratio is very different depending on whether you're working on macro or meiofauna, and some macroorganisms can move and escape from these extreme conditions. There are also complex interactions between bioturbation and the cable bacteria activity which seem to buffer this bacterial activity (Malkin et al, 2014, 2022; Aller et al, 2019). Eventually, as presented to Reviewer 1, if calcareous foraminifera are decalcified so intensely, this means that despite the strong physical and biogeochemical dynamics of this kind of transitional environment in time and space, the corrosive conditions are sufficiently strong in intensity through time to generate dissolution in living organisms that can fight off these hostile conditions to a greater or lesser extent.

P17L346 Based on the information given in the introduction, it appears that the discussion is addressing one of the objectives of the study. However, there seems to be a lack of material to support the argument. It is not possible to determine the reproducibility of the discussion about the extent of CBA, especially with the absence of CB abundance data at St. 3. For instance, during sediment sample processing, was the presence of CB confirmed visually or through other means? Or was it only left on photographs? If the presence of CB was confirmed, it would be possible to describe the amount of CB present in Table 1 qualitatively. If you have microscopic or visual observations in the form of a bacterial mat, it would be supportive. If such data is available, it would be a good idea to add it to the figure or supplement data. The presence of CB in the sediments was not confirmed visually in 2020. We recently attempted to make microscopic observations and take photographs of CB by incubating sediment from stations 2 and 3 using the method described by Thorup et al. (2021). We have been able to observe balls of filaments whose scale and structures seemed to indicate that they were indeed CB. These observations were in low abundance. This is not surprising given the calculated densities, which remain within the low range values reported in the literature (see "Discussion", lines 330-341). However, the combination of pH and oxygen microprofiles provides a very high degree of confidence that they are active and therefore present at station 3.

P18L403 I am largely in agreement with the authors' perspectives. However, I recommend a more extensive engagement with the discussions on test dissolution from prior studies as outlined in Introduction lines 61-65, and subsequently, a further elaboration on the assertion that "the influence of CBA cannot be overlooked and may be predominant in certain locales." Following these remarks and those of Reviewer 2, we plan to restructure the Discussion section. The dissolution process in Auray would be discussed in more detail based on the bibliography and the data in Marie Fouet's thesis. It would conclude with the hypothesis that the CBA seemed to be the main contributor in this case. The section titles and abstract will be modified accordingly.

We will also add a brief development of the hypothesis of dependent species response based on the work of McIntyre-Wressnig et al. (2014), Haynert et al. (2014), Charrieau et al. (2018c) and Mojtahid et al. (2023).

While the manuscript addresses the mineralization of organic carbon and the intensification of acidic environments, the discourse concerning carbonate saturation appears to be insufficient. As delineated by the authors in lines 63 and 69, test dissolution corresponds with a reduction in carbonate saturation. Carbonate saturation is functionally related to the concentrations of carbonate and/or calcium ions. Hence, it would be pertinent to include a discussion on calcium ions (even if there are no observed variations), as opposed to solely focusing on pH. The solitary mention of carbonate saturation at line 444 falls short of providing a comprehensive understanding. We share this criticism. In our view, it goes without saying that the acidification processes that coastal environments undergo, and which are discussed here, include the carbonate saturation. We can mention this more during the discussion. We are keen to remain general about the complexity of the chemical processes involved so as not to confuse the reader and to avoid making statements that would be highly speculative given the data available to us.

P20L449 I agree with the importance of the authors' perspective in incorporating the new perspective of CBA into the discussion of foraminifera distribution. We thank the reviewer for agreeing with our thesis.

P21L490 It is crucial to consider the authors' point of view. For instance, it would be beneficial to develop a proxy that can detect the existence and strength of dissolution by CBA in the future. Additionally, it might be necessary to acknowledge the potential of modifying the process of micropaleontological sediment treatment due to the assumption of shell dissolution. A multivariate approach coupling (1) the identification of lipid biomarkers in cable bacteria or eDNA and their investigation in ancient sediments to determine their presence and (2) the study of foraminiferal species assemblages (C/T ratio), shell preservation and isotopic shell composition, could be a good candidate to try out.

References

Aller et al, 2019 (10.1126/sciadv.aaw3651)

Charrieau et al. (2018c, 10.1016/j.marenvres.2018.03.015)

Charrieau et al. (2022, 10.1038/s41598-022-10375-w)

Redois (1998, PhD report, fr)

Fouet (2022, PhD report, fr/en, http://www.theses.fr/s227694)

Fouet et al. (2022, 10.3390/w14040645)

Haynert et al. (2014, 10.5194/bg-11-1581-2014)

Malkin et al, 2014 (10.1038/ismej.2014.41)

Malkin et al., 2022 (10.1002/lno.12087)

McIntyre-Wressnig et al. (2014, 10.2113/gsjfr.44.4.341)

Mojtahid et al. (2023, 10.1016/j.chemgeo.2023.121396)

Thorup et al. (2021, 10.1016/j.syapm.2021.126236)

---

## Author Comment (AC3)

**Response to Reviewer 2: anonymous (https://doi.org/10.5194/bg-2023-169-RC2)**

**General comments:**

The manuscript presents an observational study documenting the abundance of living and dead calcareous foraminifera and their dissolution condition, in conjunction with estimates of cable bacteria abundance and their associated acidification of porewaters. The study sampled 3 intertidal study sites of a macrotidal estuary on one occasion each. The data collected are of high quality (vis-à-vis the identification, quantification, and imaging of the forams by SEM and cable bacteria by qPCR and microsensor profiling), and are a valuable contribution to the literature.

Despite the rigor of the data collection, I am concerned that the study places strong emphasis on concluding the role of cable bacteria based on a limited number of samples and range of conditions (n=3). As presented by the authors, a priori, there is strong evidence to suggest that high cable bacteria activity acidifies porewaters in marine and estuarine settings more than other microbial activities, and strong theoretical evidence that the impacts of this acidity on the saturation state of calcium carbonate may drive dissolution of calcareous foraminiferal tests, with implications for their interpretation as bioindicators or paleoproxies. However, the activity and impact of cable bacteria on porewater acidity may have large variability over space and time, which remains poorly characterized, while foram test dissolution is presumably a property integrated over months or longer. We would like to clarify that experimental data show a much shorter timescale for dissolution processes in the tests of living foraminifera, of the order of a few days to a few weeks (Le Cadre et al., 2003; Charrieau et al., 2018, 2022; Daviray, pers. com.). These microorganisms are capable of recalcifying their test following acidification events with the same daily to weekly dynamics. This dynamic is relatively comparable to that of cable bacteria, as are the oxidation processes of the reduced mineral phases that can generate acidification of the sediment. We therefore assume that the shells of dead specimens incorporate the variability of these dynamics to a greater or lesser extent. This information will be added to the manuscript.

Therefore, I think the assertion that cable bacteria activity is the main driver of foram dissolution (or the singular driver, as implied in the manuscript) should be treated as far more tentative than the manuscript presents. Although I recognize the authors make an effort to identify that their conclusions that cable bacteria cause acid dissolution of foraminifera tests are tentative by including "potential" in the title, I nevertheless think some statements and the organization of the text tends to oversell the conclusive role of cable bacteria, based on the new data presented. I provide some specific suggestions below which I hope the authors will find useful.

**Specific comments:**

I think the authors should consider re-framing the narrative as foremost a report of dissolution stages of foraminiferal tests in intertidal sediments, with the examination of the role of cable bacteria as secondary. This suggestion implies reorganizing the title, Abstract, Introduction, Results, and Discussion to put the examination of the foraminifera first, and the examination of cable bacteria second. (E.g., Dissolution rates of hard shelled benthic foraminifera and potential contribution of cable bacteria activity). The opening statement of the Conclusions is similarly overstated in my opinion and should be edited (i.e., "... *strongly* suggests that sediment acidification caused by CBA could be responsible for significant calcareous test foraminifera dissolution patterns"). [emphasis mine]. We agree to the reviewer that we were sometimes too enthusiastic about our work hypothesis. Nevertheless, we believe that our choice to structure the Discussion section remains relevant by attempting to identify the factors behind the acidification process before discussing its effects. We suggest replacing in the titles of Discussion sections 4.2 and 4.3 "*Impact of cable bacteria*" with "*Impact of porewater*

*acidification*". We also consider to moderating our comments by replacing "*CBA*" with "*acidification process*" in the text body. Discussion section 4.1 will develop further other processes that could generate this acidification in these transitional environments (see response to reviewer 1). Discussion sections 4.2 and 4.3 will be slightly reorganised to develop more about implications of pH on living and dead foraminifera shell integrity and assemblages according to the literature to better bring our hypothesis to the reader as a very likely hypothesis that deserves consideration and further work by the community. Eventually, apart from the "*strongly*" which will be deleted, we believe our conclusion is not too assertive and invites the community to consider cable bacteria activity as a phenomenon that could cause misinterpretation in analysing benthic foraminifera as bio- or paleoindicators.

Line 25 states that strong and weak cable bacteria activity was associated with pH 5.8. This seems to contradict the previous statements that cable bacteria activity was assessed with pH microsensors. Please clarify. Indeed, our wording can lead to confusion. We propose to modify it as follows: "*Highly contrasting CBA (from low to very intense) were described with sediment acidification from 1.0 to 2.4 $\Delta pH$*".

(Note that qPCR quantifies cable bacteria abundance or density, which likely scales with activity, but is not strictly equivalent). We agree and are aware about it.

Technical Corrections

It is probably up to the discretion of the authors, but I recommend minimizing unnecessary acronyms, like Cable Bacteria and Cable Bacteria Activity, to improve readability. This suggestion will be considered to aid readability.

Line 20: no parentheses or hyphen required for "until 5 cm depth". We agree. They will be deleted.

lines 357 and 358: check spelling of author names These mistakes will be corrected.

References

Le Cadre et al., 2003 (10.2113/0330001)

Charrieau et al., 2018 (10.1016/j.marenvres.2018.03.015)